# Chemical Compositions and Characteristics of Biocalcium from Asian Sea Bass (*Lates calcarifer*) Scales as Influenced by Pretreatment and Heating Processes

**DOI:** 10.3390/foods12142695

**Published:** 2023-07-13

**Authors:** Krisana Nilsuwan, Saowakon Pomtong, Afeefah Chedosama, Pornsatit Sookchoo, Soottawat Benjakul

**Affiliations:** 1International Center of Excellence in Seafood Science and Innovation (ICE-SSI), Faculty of Agro-Industry, Prince of Songkla University, Hat Yai, Songkhla 90110, Thailand; krisana.n@psu.ac.th (K.N.); nampomtong43@gmail.com (S.P.); afeefah.615@gmail.com (A.C.); 2Center of Excellence in Bio-Based Materials and Packaging Innovation, Faculty of Agro-Industry, Prince of Songkla University, Hat Yai, Songkhla 90110, Thailand; pornsatit.s@psu.ac.th; 3Department of Food and Nutrition, Kyung Hee University, Seoul 02447, Republic of Korea

**Keywords:** biocalcium, Asian sea bass, scale, pretreatment, heating process

## Abstract

Asian sea bass scales discarded from the fish processing industry contain collagen and calcium. The production of biocalcium can increase their value. The effect of alkaline pretreatment on non-collagenous protein removal from scales was investigated. The alkaline pretreatment of scales was optimal when 2 M NaOH solution was used for 10 min. The impacts of heating processes of varying times on chemical compositions and characteristics of biocalcium (BC) powder from alkali-pretreated scales were also studied. A lower loss of hydroxyproline (HYP) and decreased hardness of scales were obtained when the scales were treated with a boiling process. BC powders from the scales subjected to boiling (B-BC) had higher yield and HYP content than BC powders using a high-pressure heating (HP-BC) process. An augmented heating time (10–30 min) lowered yield, HYP, moisture, and protein contents in BC powder regardless of the heating processes. HP-BC powder had higher ash, calcium, and phosphorus contents than B-BC powder. A whiter color and larger mean particle size were attained for the B-BC powders. X-ray diffractograms revealed that all BC powders had hydroxyapatite, which had a crystallinity of 53.60–66.54%, as a major component. FTIR spectra confirmed that all BC powders comprised proteins and inorganic matter. BC powder from scales with high yield and satisfactory characteristics could be used in calcium supplements.

## 1. Introduction

Calcium is an essential mineral for maintaining human health. Some diseases such as osteoporosis, the reduction of bone mass, and rickets have become global problems associated with the insufficient intake of calcium [1]. To conquer such problems, the fortification of calcium in foods is another alternative. However, some calcium salts such as calcium carbonate show low absorption into the human body [2,3]. Thus, the enhanced calcium absorption is preliminarily required [4]. Biocalcium, which is generally derived from fish bones, the leftover from fish processing or dressing, has drawn augmenting attention. Biocalcium is the calcium produced from biological materials such as egg-shells, fish bones, and fish eyeball scleral cartilage [5,6,7]. Biocalcium generally has high content of collagen, calcium, and phosphorus, which are assembled in a hydroxyapatite form [7]. Collagenous proteins associated with calcium in fish bone can increase calcium solubility and bioavailability [2,3]. Collagen peptides present during the gastro-intestinal track digestion of biocalcium could function as calcium carriers, thereby increasing calcium solubility [3]. These peptides could also induce the paracellular transportation of calcium across the cells in the small intestine, leading to the high bioavailability of biocalcium [3].

Thailand has become one of the world’s largest fishery exporters, producing around 2 million tons per year [8]. In 2018, the GDP of the Thai fishery industry was USD 3560 million [8]. The total value of fishery exports increased to USD 3.9 billion in 2017 [8]. Thai fishery industries produce and export a wide range of fishery products in many forms, e.g. raw, cooked, ready-to-eat, etc. [8]. Leftovers involving scales in large quantities are generated from the fish processing industry. The fish scales removed during the descaling process are useless and generally discarded [9]. Fish scales are constructed with collagen and minerals, especially calcium [10]. Fish scale calcium is therefore beneficial and can serve as a potential raw material for biocalcium production since it is priceless and abundant. Moreover, the utilization of fish scales can minimize environmental pollution and bring about the value-added products.

The production of biocalcium from fish processing leftovers such as fish bones has been of increasing interest [11,12]. For the production of different forms of calcium from bones containing biocalcium, hydroxyapatite, calcium oxide, etc., several factors, including pretreatment using alkaline [2,12], heating temperature, time for calcination, and bleaching agents, have been documented [7,13,14]. High strength and stiffness of scales are attributed to the high amount of minerals. Comprising inorganic particles in the vicinity of a flexible collagen matrix, a scale has high toughness in nature [10,15]. Because of its strong and tough structure, it is hard to reduce the scale to a smaller size or powder via mechanical grinding or pulverization. Wijayanti et al. [16] documented that heat treatment could be used for lowering the toughness of fish bones. In this regard, sufficient heating should be implemented to obtain softened fish scales. Nevertheless, the influence of heating processes employed for softening scales has never been documented. The purposes of this study were to investigate the influence of alkaline pretreatment on the non-collagenous protein removal of Asian sea bass scales and to elucidate the impact of different heating processes on the yield, chemical composition, and characteristics of biocalcium from Asian sea bass scales.

## 2. Materials and Methods

### 2.1. Chemicals

Sodium hydroxide pellets and isopropanol were acquired from KemAus™ (Phakanong, Bangkok, Thailand). Hydrochloric acid and sulfuric acid were obtained from QReC (Auckland, New Zealand). Petroleum ether (J. T. Baker, Leicestershire, UK) was used for analysis of fat content. All chemicals were of analytical grade.

### 2.2. Effect of Alkaline Pretreatment on Non-Collagenous Protein Removal of Asian Sea Bass Scales

Scales from Asian sea bass (*Lates calcarifer*) having the weight of 3–5 kg/fish were obtained from a local fresh market in Songkhla Province, Thailand. The scales (100 g) were mixed with sodium hydroxide (NaOH) solution at varying concentrations (0.5, 1, and 2 M) at a 1:10 (*w*/*v*) ratio. The mixtures were stirred with an overhead stirrer equipped with a propeller (RW 20 digital, IKA-Werke GmbH & CO.KG, Staufen, Germany) at the speed of 150 rpm for 120 min at room temperature (RT). Five milliliters of solution were taken at 0, 10, 20, 30, 40, 50, 60, 90, and 120 min and subjected to determination of total soluble protein content (using the Biuret method) [17] and hydroxyproline contents [18]. The condition rendering high non-collagenous protein removal while maintaining hydroxyproline in scales was chosen for further study.

### 2.3. Effect of Heating Processes on Softening of Scales

#### 2.3.1. Heat Treatment Processes

The scales were pretreated with 2 M NaOH solution at a 1:10 (*w*/*v*) ratio (10 min, RT) and washed using running tap water, thoroughly, until pH 7 was achieved. Pretreated scales (PS) (100 g) were transferred into an Erlenmeyer flask containing 2 volumes of distilled water and covered with aluminum foil. The prepared samples were divided into two groups. The first and second groups of samples were treated with boiling water (100 °C) and high-pressure heating (120 °C, 15.9 psi), respectively. Varying heating times (10, 20, and 30 min) were used. After heating, the scales were collected with the aid of a sieve (40 mesh). The filtrates were taken and the hydroxyproline contents were measured [18]. All heat-treated scales were dried in a rotary-tray dryer with an air velocity of 1.5 m/s at 50 °C for 12 h. The dried scales were then analyzed.

#### 2.3.2. Textural Property

The hardness and fracturability of scales under different heating processes and with different times were determined by using a texture analyzer (TA.XT Plus Texture Analyzer, Stable Micro Systems, Surrey, UK) as tailored by Nilsuwan et al. [19]. Five pieces (1 g) of scales with diameters of around 1 cm each were placed in the Ottawa cell. Then, a flat square probe was used for analysis. Measurement was performed with the aid of a load cell of 50 kg. A pre-test speed of 1.0 mm/s, test speed of 2.0 mm/s, and distance of 5.0 mm were used. The maximum force and slope required to break the scales were recorded.

### 2.4. Production and Characterization of Biocalcium

The dried scales obtained after non-collagenous protein removal and heating under the optimal condition were ground into coarse particles using a high-speed blender (Panasonic, Model MX-898N, Berkshire, UK). Then, fine particles were produced using a ball milling machine (Model PM 100, Retsch GmbH, Haan, Germany). Samples (200 mL) were transferred into a grinding jar containing 25 grinding balls (diameter: 20 mm). One-direction rotation mode was applied and grinding was done at a speed of 200 rpm for 2.5 h. Subsequently, the ground sample was sieved using an electric sieving machine (E.V.S.1., Endecotts Ltd., London, England) equipped with a sieve (200 mesh). The sieving was operated with a frequency of 50 Hz for 60 min. Samples having particle sizes lower than 75 μm were collected. All powder samples were subjected to analyses.

#### 2.4.1. Yield

The yield was computed using the equation shown below.
$$\mathrm{Yield}\;\left(\%\right)=\frac{\mathrm{Dry}\;\mathrm{weight}\;\mathrm{of}\;\mathrm{biocalcium}\;\mathrm{powder}\;(g)}{\mathrm{Dry}\;\mathrm{weight}\;\mathrm{of}\;\mathrm{initial}\;\mathrm{scale}\;(g)}\times 100$$

#### 2.4.2. Hydroxyproline Content

Hydroxyproline content in the samples was examined by using the spectrophotometric method as tailored by Bergman and Loxley [18].

#### 2.4.3. Chemical Composition

Biocalcium powders were analyzed for moisture, protein, fat, and ash contents using AOAC method with analytical method numbers of 925.45, 981.10, 948.15, and 923.03, respectively [20]. 

#### 2.4.4. Calcium and Phosphorus Contents

Ca and P contents in all samples were determined using an inductively coupled plasma optical emission spectrometer (ICP-OES) (Model Optima 4300 DV, Perkin Elmer, Shelton, MA, USA). The detection was done at wavelengths of 317.933 and 213.617 nm for Ca and P, respectively, following the method of Feist and Mikula [21].

#### 2.4.5. Color

Colors of samples were measured using a Hunterlab Colorfex EZ colorimeter (Hunter Associates Laboratory Inc., Reston, VA, USA). Lightness (*L**), redness/greenness (*a**), and yellowness/blueness (*b**) were reported. The differences in color (Δ*E**) between the samples and white plate standard (*L** = 92.8, *a** = −1.27, *b** = 0.35) were also computed using the equation shown below.
$$\Delta E^{*}=\sqrt{{\left(\Delta {L\;}^{*}\right)}^{2}+(\Delta {a\;}^{*}{)}^{2}+(\Delta {b\;}^{*}{)}^{2}}$$

#### 2.4.6. Mean Particle Size

Mean particle size of all powder samples was measured with the aid of a liquid particle size analyzer (LPSA) (Model LS 230, Beckman Coulter^®^, Fullerton, CA, USA) as per the method of Benjakul, Mad-Ali, Senphan, and Sookchoo [2]. Each sample was firstly dispersed in distilled water and then fed into the machine with continuous stirring. Five successive readings were done. Volume-weighted mean particle diameter (*d*_43_), representing the mean diameter of a sphere with the same volume, was determined.

#### 2.4.7. X-ray Diffraction (XRD)

The phase composition of all biocalcium powders was evaluated by X-ray diffraction (XRD) as per the method of Wijayanti, Benjakul, and Sookchoo [16] using an X-ray diffractometer (WI-RES-XRD EMPREYAN-001, Panalytical, the Netherlands) at the wavelength of 0.154 nm (Cu K-α radiation). The samples were swept at a 2θ angle ranging from 20° to 70° with a time/step of 0.27 s and step size of 0.026° at 40 kV and 30 mA. Phase identification was accomplished using a peak profile matching the standard powder diffraction data file from the International Centre for Diffraction Data (ICDD).

#### 2.4.8. Fourier Transform Infrared Spectroscopy (FTIR)

FTIR spectra of biocalcium powder samples were obtained using ATR-FTIR model Equinox 55 (Bruker, Ettlingen, Germany) following the procedure of Pudtikajorn, Sae-leaw, Yesilsu, Sookchoo, and Benjakul [7]. Spectra in the range of 400–4000 cm^−1^ with step resolutions of 4 cm^−1^ were recorded using OPUS 3.0 software (Bruker, Ettlingen, Germany).

### 2.5. Statistical Analysis

All the studies were carried out using completely randomized design (CRD). Experiments and analyses were performed in triplicate (*n* = 3). Analysis of variance (ANOVA) was performed. Differences among samples were examined using Duncan’s multiple range test at the *p* < 0.05 level. The analysis was carried out with an SPSS package (SPSS for Windows, Version 28, SPSS Inc., Chicago, IL, USA).

## 3. Results and Discussion

### 3.1. Effect of Alkaline Pretreatment on Removal of Non-Collagenous Protein from Scales

Total soluble proteins (TSP) were liberated into sodium hydroxide (NaOH) solutions at different concentrations (0.5, 1 and 2 M) as a function of time from the Asian sea bass scales (Figure 1A). Overall, the TSP content in the NaOH solutions was increased as pretreatment time upsurged (*p* < 0.05), regardless of the NaOH concentrations. At the same pretreatment time, higher TSP content (*p* < 0.05) was found in the 2 M NaOH solution used for the treatment of scales compared to the TSP contents found in NaOH solutions at lower concentrations (0.5 and 1 M) (*p* < 0.05). Within the first 10 min of pretreatment, sharp increases in extractable protein in the NaOH solutions were observed, especially when NaOH solutions at 2 M were employed. Overall, the gradual increase in TSP was noticeable for all NaOH solutions from 30 min up to 120 min. Alkali could solubilize the proteins attached to the scales or localized in the scales, leading to the removal of proteinaceous substances from the scales. Proteins likely underwent solubilization or unfolding with increasing alkaline concentration and pretreatment time. These processes were related to an increase in the mass transfer of solubilized or denatured proteins from the scales. Idowu, Benjakul, Sinthusamran, Sae-leaw, Suzuki, Kitani, and Sookchoo [12] documented that alkaline solutions effectively removed non-collagenous proteins from salmon frames used for biocalcium production. 

The content of hydroxyproline (HYP), a distinct amino acid representing collagen, was monitored during alkaline pretreatment, as shown in Figure 1B. Typically, scales consist of collagen and hydroxyapatide, which are together distributed in well-defined layers from the bottom to the top of each scale [22]. Collagenous proteins were co-solubilized with other proteins, particularly when continuous stirring was applied. Basically, native collagen is not soluble at the neutral or alkaline pH range [23]. Generally, these collagenous proteins were liberated continuously up to 120 min of pretreatment using NaOH solution. Moreover, higher HYP content (*p* < 0.05) was detected in NaOH solutions at higher concentrations when pretreatment time increased. Overall, alkaline pretreatment with 2 M NaOH for 10 min could remove proteins effectively while lowering the loss of hydroxyproline, representing collagen. This alkaline pretreatment condition was selected for the removal of non-collagenous protein from the scales.

### 3.2. Effect of Heating Processes on Softening of Scales

#### 3.2.1. HYP Release from Scales

After heating the alkali-treated scales using different processes, the HYP content of the water used as the heating medium, in which the scales were present, was determined (Figure 2). The heating processes exhibited the pronounced effect on HYP content in water. Longer heating times increased HYP content in water (*p* < 0.05), regardless of the heating processes used. At the same heating time, a higher HYP content (*p* < 0.05) was detected when high-pressure heating was adopted compared to that found in boiling water. When a protein is heated, its structure is destroyed or hydrolyzed; this process is known as thermal degradation [24]. This was mainly associated with the release of proteins or peptides from scale matrices, which led to the escalation of HYP content in pretreatment water. Wijayanti, Benjakul, and Sookchoo [16] documented that protein, α-amino group, and hydroxyproline contents in water increased for tuna bones subjected to high-pressure heating at 121 °C, particularly when the heating time was increased from 30 to 90 min. Hydroxyproline content in water indicated that the collagen in the scale matrices was leached out and dissolved in water. Glycine, proline, and hydroxyproline are the dominant amino acids in scale collagen. Chuaychan et al. [25] documented that the contents of glycine, proline, and hydroxyproline in collagen from seabass scales (*Lates calcarifer*) were 327, 108, and 85 residues/1000 residues, respectively. Thus, heating processes could disrupt protein structure, especially that of collagen in scale matrices, which serves as one of the main components of fish scale structure. Thus, heating, especially high-pressure heating, could potentially destroy the collagen–hydroxyapatite complex. This brought about a looser and weaker structure in scales, thus favoring the liberation of collagen.

#### 3.2.2. Textural Property

The textural properties of the dried scales after being treated with different heating processes, expressed as hardness and fracturability, are shown in Figure 3. Generally, the scales without heat treatment had high hardness (15.5 N) and low fracturability (2.8 N/s) values (data not shown). The interaction and ordered organization between mineral and native collagen could provide strong scale structure [26]. The hardness value was decreased after the scales were treated with the boiling and high-pressure heating processes (Figure 3A). Heat treatment might have been able to destroy and fragment the triple-helix structure of collagen, which was distributed in the scale matrices. For the same heating time, lower hardness values (*p* < 0.05) were obtained for scales subjected to boiling than that for scales treated with high-pressure heating. The boiling process might have broken the structures of the scales to some extent, which might have also resulted in softened scales. Moreover, the high energy associated with high-pressure heating plausibly destroyed and fragmented the triple-helix structure of collagen to a greater extent, in which small fragments of proteins or peptides could leach out into water as indicated by the higher hydroxyproline content detected in water (Figure 2). In this context, higher interaction between inorganic substances with less protein might have been enhanced in the resulting scales that had been treated with high-pressure heating. This might lead to higher hardness than that of scales treated with boiling water. Additionally, the lowest hardness value was found in the scales after heating for 30 min, regardless of the heating processes used. Wijayanti, Benjakul, and Sookchoo [16] documented that increasing the heating time of high-pressure heating up to 90 min could effectively decrease the hardness value of Asian sea bass bone. Furthermore, continuous increases in fracturability (*p* < 0.05) were found for the scales subjected to boiling and high-pressure heating as the heating times increased (Figure 3B). Fracturability is the tendency of a material to fracture, crumble, crack, or shatter when force is applied [27]. Generally, higher fracturability was observed in the scales subjected to high-pressure heating than in those subjected to boiling. The result was in line with that for hardness (Figure 3A). This result suggested that high-pressure heating could provide brittle scales. Nevertheless, no difference in fracturability values (*p* > 0.05) was attained for the scales pretreated with boiling water for up to 30 min. Therefore, heating process showed the profound impact on the properties of scales, particularly on the softening or fracturability of the scales prior to grinding to obtain biocalcium.

### 3.3. Yield, Composition, and Characteristics of Biocalcium

#### 3.3.1. Yield

The yields of biocalcium (BC) powders produced from the scales subjected to different heating processes are shown in Table 1. The BC powders from the scales without heat treatment typically had low yields (33.1%) (data not shown). These yields were associated with the high toughness of the original scales, which could not be ground into powder completely and remained as large residues. All BC powders from the scales treated with both heating processes for 10, 20, and 30 min had yields in the range of 58.56–85.47%. The BC powders from the scales subjected to boiling exhibited higher yields (*p* < 0.05) than the BC powders from scales with high-pressure heating, irrespective of the heating times used. This result was plausibly attributed to higher retained protein in the scales after the boiling process, as indicated by their lower hydroxyproline content in treated water (as shown in Figure 2). Moreover, the yields of the BC powders from the scales under the boiling and high-pressure heating processes were decreased from 85.47% to 72.87% and from 68.53% to 60.33% when the heating time was increased from 10 to 30 min, respectively. Longer heating time caused a loss of solid fraction from the scales to a greater extent, resulting in a lower yield of BC powders. Thus, the heating process used had an influence on the yield of biocalcium. The boiling of scales for 10 min could render the biocalcium with the highest yield, and it was therefore considered a potential heating process for the production of biocalcium from Asian sea bass scales.

#### 3.3.2. Hydroxyproline Content

The contents of hydroxyproline (HYP) in the BC powders prepared from the dried scales after different heat treatments are shown in Table 1. The HYP content in the BC powders from the scales subjected to boiling was in the range of 39.66–41.47 mg/g per dry sample, irrespective of the heating time used. The BC powders prepared from the scales using high-pressure heating had lower hydroxyproline content (*p* < 0.05) than the BC powders from the scales subjected to boiling for all heating times. High-pressure heating could have enhanced the liberation of collagen from the scales, as shown by the lower hydroxyproline content in the resulting BC powders. This result was coincidental with the higher HYP content found in the water containing the scales treated using high-pressure heating, as shown in Figure 2. Wijayanti, Benjakul, and Sookchoo [14] found that the hydroxyproline contents of biocalcium from Asian sea bass backbones subjected to boiling followed by the autoclaving process were lower than those of biocalcium from Asian sea bass backbones treated with boiling water alone. Furthermore, a decrease in HYP content was generally observed when the heating time upsurged (*p* < 0.05). Nonetheless, similar HYP contents were noted between the BC powders from the scales treated for 20 and 30 min (*p* > 0.05) using both heating processes. This indicated that longer heating time caused the loss of collagen from the scales to a greater extent. Therefore, HYP content was influenced by the heat treatment processes, in which the boiling process could provide higher levels of HYP content in the resulting BC powders.

#### 3.3.3. Chemical Composition

All BC powders contained moisture, protein, fat, and ash contents in the range of 4.60–7.37, 27.82–51.35, 0.21–0.29, and 43.61–69.69%, respectively (Table 1). Benjakul, Mad-Ali, Senphan, and Sookchoo [2] reported that biocalcium from pre-cooked skipjack tuna bones had 24.26% protein, 0.21% fat, and 72.20% ash. Pudtikajorn, Sae-leaw, Yesilsu, Sookchoo, and Benjakul [7] also documented that BC powder from skipjack tuna (*Katsuwonus pelamis*) eyeball scleral cartilage had protein, fat, and ash contents of 31.33, 0.10, and 60.51%, respectively. Moreover, with different heating processes, the BC powders prepared from the scales subjected to the boiling process showed higher moisture (6.63–7.37%) and protein content (48.95–51.35%) than the BC powders prepared from the scales using high-pressure heating (4.60–5.45% and 27.82–39.98%, respectively). Mild heat treatment could retain organic compounds, especially collagen, in the scales, resulting in higher concentrations of organic matter in the scale matrices. However, no differences in fat content (*p* > 0.05) between BC powders prepared from the scales with the boiling and high-pressure heating processes were observed, regardless of the heating times. Additionally, the BC powders prepared from the scales using high-pressure heating had higher contents of ash (57.59–69.69%) compared to the BC powders prepared from the scales using the boiling process (43.61–48.17%). The high-pressure heating process could therefore reduce protein content by eliminating proteins, especially collagen, in the sample matrix [16]. This coincided well with the augmented hydroxyproline contents in the water used as the heating medium (Figure 2), whereas higher levels of inorganic content were found in the resulting BC powder. The highest ash content was noted for the BC powders prepared from the scales using high-pressure heating for 30 min (69.69%) (*p* < 0.05). Therefore, the chemical compositions of the BC powders varied depending on the heating processes used.

#### 3.3.4. Calcium and Phosphorus Contents

The Ca and P contents of all the BC powders were different among the samples (*p* < 0.05) (Table 1). The BC powders prepared from the scales treated with high-pressure heating had higher Ca and P contents (*p* < 0.05) than the BC powders prepared from the scales subjected to the boiling process. This result was consistent with the higher ash content in the former, as shown in Table 1. Lower calcium and phosphorus contents (*p* < 0.05) were found in the BC powders prepared from the scales after heating using the boiling process, compared to those in the BC powders prepared from the scales using the high-pressure heating process, regardless of the heating times used. Since the scales prepared using the boiling process still had high quantities of retained organic compounds, especially collagen, this resulted in a lower proportion of minerals but higher content of protein. The high-pressure heating process more likely caused the leaching of organic compounds from the scales, thus increasing the proportion of inorganic substances. In addition, the highest calcium and phosphorus contents were found in the BC powders from the scales subjected to the high-pressure heating process for 30 min (*p* < 0.05). High-pressure heating for a longer time could enhance the disruption of peptide bonds in the matrices of scales and release collagenous proteins and other organic compounds. This resulted in the increased concentration of minerals in the scales. Inorganic compounds such as minerals are generally stable against high temperature while organic compounds such as proteins and lipids are sensitive to heat [16].

The mole ratios of calcium and phosphorus (Ca/P) were in the ranges of 1.17–1.23 and 1.24–1.25 in the BC powders prepared using the scales subjected to the boiling and high-pressure heating (HP) processes, respectively. Typically, hydroxyapatite (HAP) has been reported as the major mineral phase in fish scales [28,29]. Pure hydroxyapatite generally contains 39.8% calcium and 18.5% phosphorus (by weight), resulting in a Ca/P mole ratio of 1.67 [30]. However, the Ca/P mole ratios of all the BC powders prepared from the Asian sea bass scales in the present study were lower than 1.67 (indicative of HAP). This was in line with the findings of de Vrieze, Heijnen, Metz, and Flik [28], who documented that Ca/P ratios in the range of 1.00–1.40 were found in zebrafish scales, relating to the stages or ages of the scales [28]. Additionally, for the same heating time, a higher mole ratio was attained for the BC powders prepared from the scales with the high-pressure heating process (1.24–1.25) compared to the powders prepared from the scales subjected to the boiling process (1.17–1.23). The increasing Ca/P mole ratio for the bones might have been due to a greater reduction in P content than in Ca content during the high-pressure heating process. Wijayanti, Benjakul, and Sookchoo [14] documented that the Ca/P mole ratio in biocalcium from boiled Asian sea bass backbones (1.64) was slightly lower than in biocalcium from autoclaved Asian sea bass backbones (1.66). The heating process with a higher energy input might enhance the co-elution of P together with organic substances, leading to a higher relative content of HAP in the resulting BC powders. Increases in the Ca/P ratios of biological apatites might be also associated with altered crystalline phases or the purity of fish scales [28,31]. However, the impact of heating time on the Ca/P ratio in the resulting BC powders was still not clear. As the heating time upsurged from 10 to 30 min, a decrease and an increase in the Ca/P ratio were observed for the BC powders prepared from the scales subjected to boiling and high-pressure heating, respectively. Nevertheless, the Ca/P mole ratios of all the BC powders prepared from the scales was lower than the Ca/P mole ratio of the BC powder generated from fish bones (1.29–1.62) [16,32]. It could be inferred that the amounts of calcium and phosphorus, as well as those of other organic matters, in bone and scale were different.

#### 3.3.5. Color

The BC powders from the scales prepared using the boiling process (B-BC) had higher values of lightness and redness (*L**- and *a**-values), along with lower values of yellowness and total difference in color (*b**- and Δ*E**-values) (*p* < 0.05), than the BC powders prepared from the scales using the high-pressure heating process (HP-BC), regardless of the heating time used (Table 2). This result indicated that the slightly higher creamy-whitish color of the HP-BC powders could be a result of the Maillard reaction. During the high-pressure heating and drying processes, protein degradation and the oxidation of remaining lipids could occur to a greater extent, leading to the formation of high free amino acids or peptides or degraded proteins and carbonyl compounds, which underwent the Maillard reaction [33]. Additionally, this phenomenon was more pronounced when longer heating times for high-pressure heating were used. This led to the augmented yellow color in the HP-BC powders, as shown by higher positive *b**-values. Nevertheless, the increasing of the heating time of the boiling process to more than 20 min decreased the *b**-values of the resulting B-BC powders. This might have been related to lower protein degradation induced by the boiling process than that of high-pressure heating counterpart. Therefore, the boiling and high-pressure heating processes determined the colors of the resulting BC powders differently, in which whiter and lower-yellowish BC powders were gained via the treatment of the scales using the boiling process.

#### 3.3.6. Particle Size

All BC powders showed different mean particle sizes, which ranged from 13.77 to 42.50 μm. The BC powders from the scales treated with high-pressure heating displayed smaller particle sizes than the BC powders from the scales subjected to the boiling process (*p* < 0.05) (Table 2) when the same heating time was used. For the same heating process used, a longer heating time resulted in a further decrease in particle size (*p* < 0.05). This result indicated that the heating process with a higher energy input for a longer time could soften the scales, which could be easily ground into particles of smaller sizes. This reconfirmed that the existence and integrity of collagen might play an important role in the stickiness and agglomeration of particles in powders. With longer heating times, collagen was much more removed. Idowu, Benjakul, Sinthusamran, Sae-leaw, Suzuki, Kitani, and Sookchoo [12] documented that the removal of collagen from salmon frame bone matrices resulted in weakened calcined bones, which could be ground into smaller sizes with ease. Particle size is the most significant factor influencing the properties and sensations of food products, involving appearance, texture, and aroma [34]. Pudtikajorn et al. [35] found that the particle size of biocalcium from skipjack tuna (*Katsuwonus pelamis*) eyeball scleral cartilage directly affected the mouthfeel of fortified products. This mouthfeel is related to acceptance or rejection by consumers.

#### 3.3.7. X-ray Diffraction (XRD) Diffractograms

The X-ray diffractograms of the BC powders from the heat-treated scales using various methods and times are shown in Figure 4. All BC powders showed diffraction peaks, following the crystalline phase of HAP (ICDD: 01-074-4172) at angles 25.95°, 31.69°, 33.07°, 39.85°, 46.66°, and 64.15°. Additionally, all BC powders showed relatively broad peaks as a result of both the elastic and the inelastic scattering of HAP nanocrystals [36]. Lower intensities with broader peaks were observed for the BC powders prepared from the scales using the boiling process, plausibly owing to the extracellular matrix and other proteins, especially collagen, in the powders [16]. On the other hand, more intense and narrower full width at half maximum (FWHM) in diffraction peaks were observed in the BC powders when the scales were subjected to high-pressure heating, particularly for longer heating times. This suggested the agglomeration of nano-hydroxyapatite to the larger crystals. A heating process with high energy input for a longer time could remove some organic substances from the scale matrices to a greater extent, thus allowing HAP crystal agglomeration to take place at a higher degree as indicated by higher crystallinity. The crystallinities of all the BC powders were in the range of 53.60–66.54%. Higher crystallinities were found for the BC powders prepared from the scales processed with high-pressure heating. For the same heating process used, increases in crystallinity were obtained in the BC powders when longer heating times were employed. This result suggested that higher crystallinity and phase purity in BC powder were related to a well-crystallized HAP phase when a heating process with a higher energy input for a longer time was applied.

#### 3.3.8. Fourier Transform Infrared Spectroscopic (FTIR) Spectra

The FTIR spectra of the BC powders from the scales treated with different heating processes and for different times are depicted in Figure 5. The characteristic peaks representing the functional groups or the bonding of all the BC powders appeared at different wavenumbers. The free N–H stretching vibration and the existence of hydrogen bonds were found at 3292 cm^−1^ (Amide A). The symmetric stretching vibration of CH_2_ at 2943 cm^−1^ (Amide B), Amide I in the range of 1635–1643 cm^−1^, Amide II in the range of 1533–1527 cm^−1^, and Amide III in the range of 1249–1239 cm^−1^ reflected the presence of proteins [7]. Benjakul, Mad-Ali, Senphan, and Sookchoo [2] documented that the absorption peaks at 1633 cm^−1^ and 1550 cm^−1^ in biocalcium were amide I and amide II, respectively, which were typical for the coiled structure of collagen. The C−O stretching vibration of carbonate (CO_3_) in the range of 1447–1449 and at 1411 cm^−1^, P−O stretching vibration of phosphate (PO_4_) in the range of 1023–1032 cm^−1^, C−O bending vibration of CO_3_ at 872 cm^−1^, and P–O bending vibration of PO_4_ at 601 and 561–562 cm^−1^ indicated the presence of minerals in the BC powders [7]. The dominant peaks of PO_4_, which were related to HAP or other inorganic substances, were observed for all sample powders at 1023, 601, and 561 cm^−1^ [37]. Moreover, higher wavenumbers, along with lower peak intensities of the Amide I, II and III bands, were generally attained for the BC powders prepared from the scales subjected to high-pressure heating process, compared to the BC powders prepared from the scales treated using the boiling process. It was noted that the thermal degradation of protein caused the loss of molecular interaction in the collagen matrices. This phenomenon was more pronounced when the high-pressure heating process was used. Wijayanti, Benjakul, and Sookchoo [16] also documented that biocalcium from autoclaved Asian sea bass backbones had lower peak intensities of Amide I and II than biocalcium from boiled Asian sea bass backbones. Additionally, gradual decreases in the peak intensities of the Amide A, B, I, II, and III bands, coincidental with the increase in the peak intensities at the wavenumbers of 1023, 872, 601 and 561 cm^−1^, were generally observed as the heating time upsurged (Figure 5B–E), regardless of the heating process used. This result indicated that the thermal degradation and denaturation of proteins, especially collagen, could be enhanced by a longer time of heating. This was related to the decreases in protein and hydroxyproline contents, which were coincidental with higher ash, Ca, and P contents as shown in Table 1.

## 4. Conclusions

Alkaline pretreatment, and the heating process used for preparing biocalcium (BC) from Asian sea bass scales, were crucial and affected both the yields and the characteristics of the prepared BC powders. Pretreatment using 2 M NaOH solution for 10 min was appropriate for the removal of non-collagenous protein from the scales. Heat treatment played a profound role in scale softening for further grinding. Higher calcium and phosphorus contents, as well as finer particle sizes, were attained for the BC powders prepared from the scales subjected to high-pressure heating, particularly when a longer heating time (30 min) was used. Therefore, BC powders from Asian sea bass scales with satisfactory chemical compositions and characteristics could be prepared by using the high-pressure heating process for 30 min after proper alkaline pretreatment.

## Figures and Tables

**Figure 1 foods-12-02695-f001:**
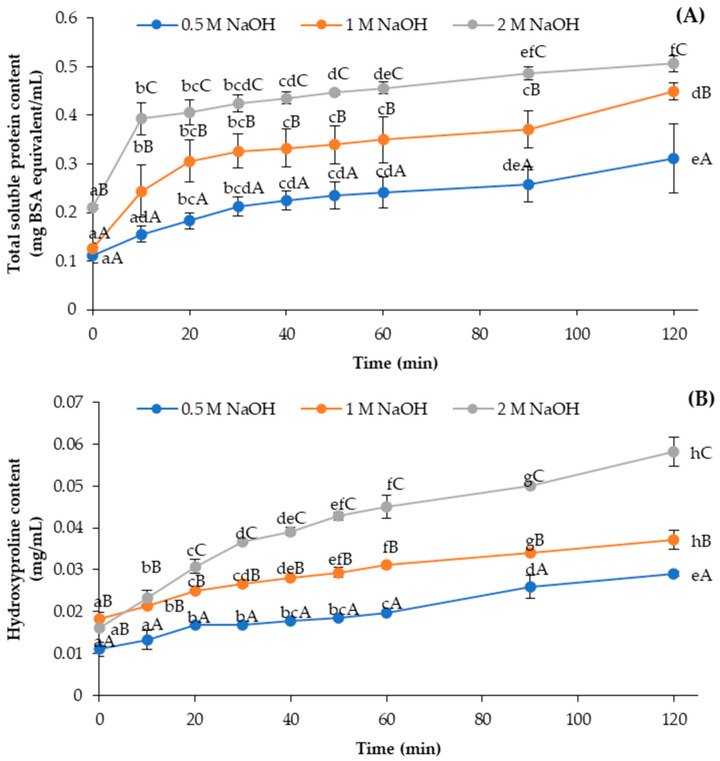
Total soluble protein (**A**) and hydroxyproline (**B**) contents in alkaline solution as functions of time during pretreatment of Asian sea bass scales using alkaline solution at different concentrations. Bars represent standard deviations (*n* = 3). Different lowercase letters within the same alkaline concentration indicate significant difference (*p* < 0.05). Different uppercase letters within the same pretreatment time indicate significant difference (*p* < 0.05).

**Figure 2 foods-12-02695-f002:**
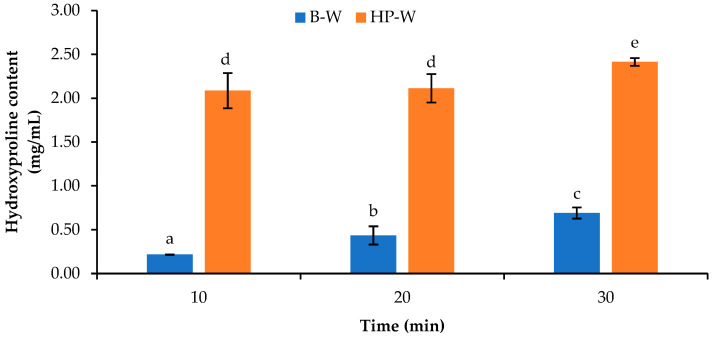
Hydroxyproline content in water used for heat treatment of Asian sea bass scales subjected to different heating processes and times. Different lowercase letters on the bars indicate significant differences (*p* < 0.05). B-W: Water used for treatment of Asian sea bass scales by boiling process; HP-W: Water used for treatment of Asian sea bass scales by high-pressure heating process. Bars represent standard deviations (*n* = 3).

**Figure 3 foods-12-02695-f003:**
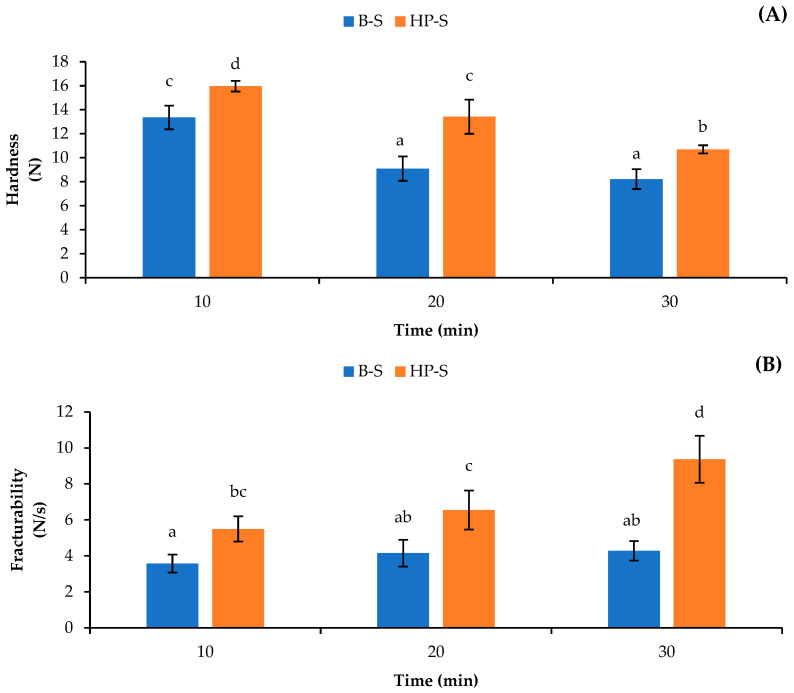
Hardness (**A**) and fracturability (**B**) of dried seabass scales after different heating processes for various times. Different lowercase letters on the bars indicate significant differences (*p* < 0.05). B-S: Asian sea bass scales subjected to boiling; HP-S: Asian sea bass scales with high-pressure heating. Bars represent the standard deviations (*n* = 3).

**Figure 4 foods-12-02695-f004:**
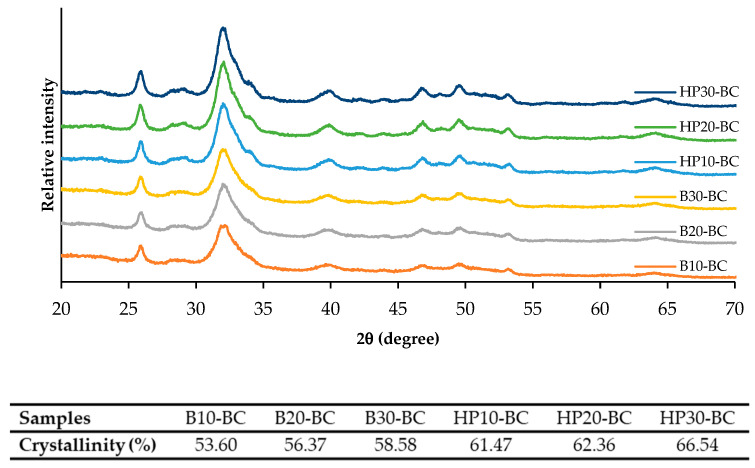
X-ray diffraction (XRD) spectra and crystallinities of biocalcium powders from Asian sea bass scales subjected to different heating processes and times. Caption: see Table 1.

**Figure 5 foods-12-02695-f005:**
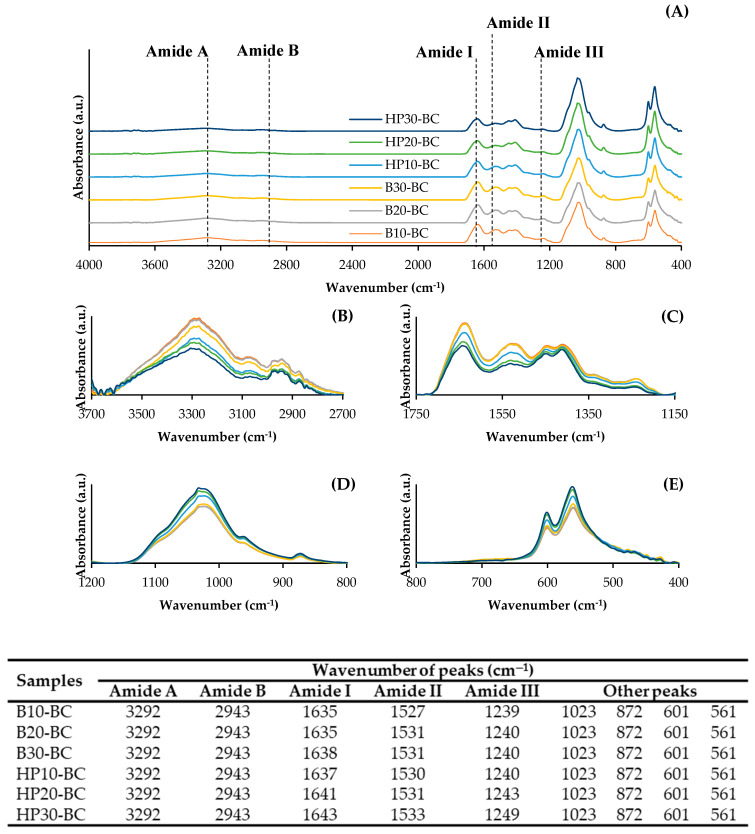
FTIR spectra at wavenumbers of 400–4000 cm^−1^ (**A**), 2700–3700 cm^−1^ (**B**), 1150–1750 cm^−1^ (**C**), 800–1200 cm^−1^ (**D**) and 400–800 cm^−1^ (**E**) for biocalcium powders prepared from Asian sea bass scales subjected to different heating processes and times. Caption: see Table 1.

**Table 1 foods-12-02695-t001:** Yield and chemical compositions of biocalcium (BC) powders from Asian sea bass scales subjected to different heat treatment processes and times.

Parameters	Samples					
	B10-BC	B20-BC	B30-BC	HP10-BC	HP20-BC	HP30-BC
Yield (%, dry weight basis)	85.47 ± 0.05 * c	83.74 ± 5.19 c	72.87 ± 3.75 b	68.53 ± 1.36 b	58.56 ± 3.26 a	60.33 ± 1.46 a
Hydroxyproline content (mg/g dry sample)	42.86 ± 0.67 d	39.58 ± 0.43 c	38.92 ± 0.34 c	31.93 ± 0.57 b	27.93 ± 0.42 a	26.52 ± 0.16 a
Moisture content (%, wet weight basis)	7.37 ± 0.22 d	6.38 ± 0.18 c	6.63 ± 0.26 c	5.45 ± 0.35 b	4.83 ± 0.26 a	4.60 ± 0.04 a
Protein content (%, dry weight basis)	51.35 ± 0.19 f	50.41 ± 1.03 f	48.95 ± 0.94 e	39.98 ± 0.93 c	35.88 ± 1.33 b	27.82 ± 0.09 a
Fat content (%, dry weight basis)	0.28 ± 0.02 b	0.22 ± 0.01 a	0.23 ± 0.02 a	0.29 ± 0.02 b	0.21 ± 0.01 a	0.21 ± 0.01 a
Ash content (%, dry weight basis)	43.61 ± 0.85 a	46.72 ± 0.4 b	48.17 ± 0.07 c	57.59 ± 0.34 d	67.37 ± 0.14 e	69.69 ± 0.14 f
Calcium (Ca) content (%, dry weight basis)	17.09 ± 0.16 a	17.84 ± 0.28 b	18.37 ± 0.18 c	23.57 ± 0.44 d	25.36 ± 0.88 e	26.13 ± 0.73 e
Phosphorus (P) content (%, dry weight basis)	10.75 ± 0.21 a	11.21 ± 0.06 b	12.18 ± 0.05 c	14.67 ± 0.10 d	15.85 ± 0.25 e	16.16 ± 0.26 f
Ca/P mole ratio	1.23	1.23	1.17	1.24	1.24	1.25

* Values are shown as mean ± SD (*n* = 3). Different lowercase letters in the same row indicate significant differences (*p* < 0.05). B10-BC, B20-BC, and B30-BC: BC powders from Asian sea bass scales treated with boiling water for 10, 20, and 30 min, respectively; HP10-BC, HP20-BC, and HP30-BC: BC powders from Asian sea bass scales subjected to high-pressure heating for 10, 20, and 30 min, respectively.

**Table 2 foods-12-02695-t002:** Colors and mean particle size diameters of biocalcium (BC) powders from Asian sea bass scales subjected to different heat treatment processes and times.

Parameters	Samples					
	B10-BC	B20-BC	B30-BC	HP10-BC	HP20-BC	HP30-BC
*L**	95.77 ± 0.12 * b	96.59 ± 0.10 e	96.42 ± 0.07 d	95.94 ± 0.11 c	95.72 ± 0.05 b	95.47 ± 0.09 a
*a**	-0.25 ± 0.04 c	-0.34 ± 0.06 c	-0.27 ± 0.02 c	-0.42 ± 0.08 b	-0.30 ± 0.08 c	-0.51 ± 0.03 a
*b**	5.27 ± 0.08 c	4.47 ± 0.09 a	4.90 ± 0.09 b	5.47 ± 0.11 c	5.91 ± 0.25 d	6.70 ± 0.07 e
Δ*E**	5.83 ± 0.01 ab	5.68 ± 0.08 a	5.90 ± 0.06 ab	6.06 ± 0.07 c	6.36 ± 0.22 d	6.92 ± 0.04 e
Mean particle size (μm)	42.50 ± 0.87 g	35.57 ± 1.23 f	32.50 ± 1.01 e	24.63 ± 0.42 d	16.07 ± 0.65 b	13.77 ± 0.25 a

* Values are shown as mean ± SD (*n* = 3). Different lowercase letters in the same row indicate significant differences (*p* < 0.05). B10-BC, B20-BC, and B30-BC: BC powder from Asian sea bass scales treated with boiling water for 10, 20, and 30 min, respectively; HP10-BC, HP20-BC, and HP30-BC: BC powder from Asian sea bass scales subjected to high-pressure heating for 10, 20, and 30 min, respectively.

## Data Availability

The data used to support the findings of this study can be made available by the corresponding author upon request.

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
