# Peer review of "Chemical Compositions and Characteristics of Biocalcium from Asian Sea Bass (Lates calcarifer) Scales as Influenced by Pretreatment and Heating Processes"

_foods, 2023, doi:10.3390/foods12142695_

Round 1
Reviewer 1 Report
The study could attract those engaged in fish by-products, mainly those working to obtain compounds with biological properties. The methodology was described correctly, and the results were discussed appropriately. However, it requires to be attended some points to give a more attractive manuscript:
The term "biocalcium" must be justified because the authors did not make any biological evaluation; they made a calcium extraction and chemical-structural characterization of the obtained compound.
P1L43-44 and P2L45-48: Because the authors give relevant information and justify why to work with fish scales, adding the newest statistical data reported by an official organism as the Yearbook of fisheries statistics of Thailand is essential. The references employed were published between 2015-2018. Moreover, on P1L44-P2L45, the authors state: "The total value of fishery exports increased from 5.6 billion US dollars in 2016 to 5.8 billion US dollars in 2017. Thai fishery industries manufacture and export a variety of fishery products such as frozen products, semi-processed and value-added products.", and the cited references were published in 2016 (Ref 5) and 2015 (Ref 6).
Please review P2L54-55: Try to employ references from 2019-2023 or rewrite the sentence.
Please modify Table 1; it is hard to understand—suggestion adding a new table (Table 2) that reports color and particle size.
Main observations on a discussion of results,
- Please compare your results with the newest data.
- Although the influence of the heating process employed for softening the scale has yet to be documented, the obtained calcium yield could be compared with similar studies.
Author Response
Response to reviewer
Comments and Suggestions for Authors
The study could attract those engaged in fish by-products, mainly those working to obtain compounds with biological properties. The methodology was described correctly, and the results were discussed appropriately. However, it requires to be attended some points to give a more attractive manuscript:
***Thank you for your kind understanding. The corrections following the reviewer’s suggestions have been provided as highlighted in yellow.
- The term "biocalcium" must be justified because the authors did not make any biological evaluation; they made a calcium extraction and chemical-structural characterization of the obtained compound.
*** Thank you very much for the insightful suggestion. The definition of “Biocalcium has been provided for better understanding along with the references. Biocalcium is the calcium produced from biological materials such as egg-shell, fish bone, and fish eyeball scleral cartilage [5-7]. Biocalcium generally has high content of collagen, calcium and phosphorus, which are assembled in a hydroxyapatite form [7].”. The aforementioned information has been provided in text. Please see line 37 – 40 and 529 – 534.
References
[5] Hassan, N.M.M. Chicken eggshell powder as dietary calcium source in biscuits. World J. Dairy Food Sci 2015, 10, 199-206.
[6] Wijayanti, I.; Prodpran, T.; Sookchoo, P.; Nirmal, N.; Zhang, B.; Balange, A.; Benjakul, S. Textural, rheological and sensorial properties of mayonnaise fortified with Asian sea bass bio-calcium. J. Am. Oil Chem. Soc 2023, 100, 123-140, doi:https://doi.org/10.1002/aocs.12649.
[7] Pudtikajorn, K.; Sae-leaw, T.; Yesilsu, A.F.; Sookchoo, P.; Benjakul, S. Process development and characteristics of biocalcium from skipjack tuna (Katsuwonus pelamis) eyeball scleral cartilage. Waste Biomass Valori. 2023, doi:10.1007/s12649-023-02075-x.
- P1L43-44 and P2L45-48: Because the authors give relevant information and justify why to work with fish scales, adding the newest statistical data reported by an official organism as the Yearbook of fisheries statistics of Thailand is essential. The references employed were published between 2015-2018. Moreover, on P1L44-P2L45, the authors state: "The total value of fishery exports increased from 5.6 billion US dollars in 2016 to 5.8 billion US dollars in 2017. Thai fishery industries manufacture and export a variety of fishery products such as frozen products, semi-processed and value-added products.", and the cited references were published in 2016 (Ref 5) and 2015 (Ref 6).
***Thank you for valuable comment and up-to-date information on fisheries statistics of Thailand. The sentences have been improved and the relevant information has been updated together with the references. Please see line 46 – 51 and 535 – 536.
References
[8] Wongrak, G.; Hur, N.; Pyo, I.; Kim, J. The impact of the EU IUU regulation on the sustainability of the Thai fishing industry. Sustainability 2021, 13, doi:10.3390/su13126814.
- Please review P2L54-55: Try to employ references from 2019-2023 or rewrite the sentence.
***Thank you for suggestion. The update references have been provided. Please see line 59 and 542 – 547.
References
[11] Wijayanti, I.; Benjakul, S.; Chantakun, K.; Prodpran, T.; Sookchoo, P. Effect of Asian sea bass bio-calcium on textural, rheological, sensorial properties and nutritive value of Indian mackerel fish spread at different levels of potato starch. Int. J. Food Sci. Technol. 2022, 57, 3181-3195, doi:https://doi.org/10.1111/ijfs.15651.
[12] Idowu, A.T.; Benjakul, S.; Sinthusamran, S.; Sae-leaw, T.; Suzuki, N.; Kitani, Y.; Sookchoo, P. Effect of alkaline treatment on characteristics of bio-calcium and hydroxyapatite powders derived from salmon bone. Applied Sciences 2020, 10, 4141, doi:10.3390/app10124141.
- Please modify Table 1; it is hard to understand—suggestion adding a new table (Table 2) that reports color and particle size.
***Thank you very much for your kind suggestion. A new table (Table 2) containing color and particle size has been added and its citation in text has also been updated. Please see line 293 – 296, 388 – 394, 399 and 417.
- Main observations on a discussion of results, please compare your results with the newest data. Although the influence of the heating process employed for softening the scale has yet to be documented, the obtained calcium yield could be compared with similar studies.
***We do agree with the reviewer’s comment. However, as the reviewer mentioned, the information of biocalcium from fish scale has not been documented. The obtained calcium yield in the present study has not been compared with the other works. The comment is well taken into consideration for our future work. In fact, the study on biocalcium from other fish scales is on-going in our lab, which will be reported in other manuscript.
However, the comparison between yields of biocalcium powder obtained from different heating processes had already been discussed, which can be useful information for the reader. Please see line 278 – 287.

Reviewer 2 Report
This manuscript is about chemical compositions and characteristics of biocalcium from Asian sea bass (Lates Calcarifer) scale as influenced by pre-treatment and heating processes. It is interesting and I think, after major revision of manuscript. You can find my comments in below:
1. The manuscript must be revised grammatically and the English level of it must be improved by a native editor.
2. The authors must re-write the abstract and conclusion sections. I think some sentences are not needed to be in these sections.
3. In line 42, please add more details about the mechanism of increasing calcium solubility and bioavailability.
4. In introduction section, please add more references in the last paragraph of this section.
5. Section 2.1 needs to be completely revised. Give more details.
6. Why did the authors select alkaline and heating process as pre-treatments? Why others not?
7. Give more details about the production of biocalcium (section 2.4).
8. In section 3.2.2, why did the authors only mention hardness and fractureability?
9. In results and discussion section, it is better to compare the obtained results with previous researches. Please do this in all parts of this section.
10. Please increase the DPI values of figures. The quality of them is poor. Also, please make the tables more desirable. They are not looking that much good.
The manuscript must be revised grammatically and the English level of it must be improved by a native editor.
Author Response
Response to reviewer
Comments and Suggestions for Authors
This manuscript is about chemical compositions and characteristics of biocalcium from Asian sea bass (Lates Calcarifer) scale as influenced by pre-treatment and heating processes. It is interesting and I think, after major revision of manuscript. You can find my comments in below:
***Thank you for valuable comments and suggestions. All queries have been responded and the corrections as per the reviewer’s comments and suggestions have been provided in text as highlighted in green.
- The manuscript must be revised grammatically and the English level of it must be improved by a native editor.
***The grammar has been checked throughout the text by a native speaker and the software ‘Grammarly’ has been used for cross-checking.. Thank you for suggestion.
- The authors must re-write the abstract and conclusion sections. I think some sentences are not needed to be in these sections.
***The abstract had been written following the guideline of Foods, which provided the brief of background, methods, results and conclusion. All sentences are necessary and related with the present study. This is a better way to provide the relevant information to the reader. We would like to keep all sentences in the abstract. However, some sentences in conclusion have been modified to make the conclusion more concise. Please see line 495 – 502.
- In line 42, please add more details about the mechanism of increasing calcium solubility and bioavailability.
***More details and related references have been provided in text. Please see line 42 – 45.
- In introduction section, please add more references in the last paragraph of this section.
***More references have been added following the reviewer’s suggestion. Please see line 62, 65 and 553 – 557.
References
[15] Qin, D.; Bi, S.; You, X.; Wang, M.; Cong, X.; Yuan, C.; Yu, M.; Cheng, X.; Chen, X.-G. Development and application of fish scale wastes as versatile natural biomaterials. Chemical Engineering Journal 2022, 428, 131102, doi:https://doi.org/10.1016/j.cej.2021.131102.
[16] Wijayanti, I.; Benjakul, S.; Sookchoo, P. Preheat-treatment and bleaching agents affect characteristics of bio-calcium from Asian sea bass (Lates calcarifer) backbone. Waste Biomass Valori. 2021, 12, 3371-3382, doi:10.1007/s12649-020-01224-w.
- Section 2.1 needs to be completely revised. Give more details.
***Thank you very much for comment. More details have been provided. Please see line 76 – 79.
- Why did the authors select alkaline and heating process as pre-treatments? Why others not?
***Alkaline solution is well-known for protein solubilization (Idowu, A.T.; Benjakul, S.; Sinthusamran, S.; Sae-leaw, T.; Suzuki, N.; Kitani, Y.; Sookchoo, P. Effect of alkaline treatment on characteristics of bio-calcium and hydroxyapatite powders derived from salmon bone. Applied Sciences 2020, 10, 4141, doi:10.3390/app10124141.). It has been widely used for removal of non-collagenous proteins from the scale. Therefore, the optimization of alkaline pretreatment had been investigated in the present study. Also, collagen is not soluble at alkaline pH range. However, it can be leached out along with other proteins, especially during agitation. Thus, the optimization must be performed to lower the loss of collagen, while enhancing the elimination of other proteins.
Additionally, because of strong and tough structure, it is hard to reduce the scale to smaller size or powder. Based on the study of our research group, Wijayanti, et al. [17] documented that heat treatment could be used for lowering the toughness of fish bone. For better clarity, the rationale had already been mentioned in the introduction section. Please see line 66 – 69.
References
[17] Wijayanti, I.; Benjakul, S.; Sookchoo, P. Effect of high pressure heating on physical and chemical characteristics of Asian sea bass (Lates calcarifer) backbone. J. Food Sci. Technol. 2021, 58, 3120-3129, doi:10.1007/s13197-020-04815-6.
- Give more details about the production of biocalcium (section 2.4).
***All details about the production of biocalcium had been mentioned in section 2.4. Please see line 113 – 123.
- In section 3.2.2, why did the authors only mention hardness and fractureability?
***Because the hardness and fractuability could provide scientific information how the scale was softened by heating treatment and how the scale could be ground or fractured into powder, respectively.
- In results and discussion section, it is better to compare the obtained results with previous researches. Please do this in all parts of this section.
***There is no information related to the influence of heating process employed for softening the scale for production of biocalcium has been documented. They focused on different aspects such as:
- Synthesis of hydroxyapatite from Labeo rohita fish scale for biomedical application (Deb, P.; Barua, E.; Das Lala, S.; Deoghare, A.B. Synthesis of hydroxyapatite from Labeo rohita fish scale for biomedical application. Materials Today: Proceedings 2019, 15, 277-283, doi:https://doi.org/10.1016/j.matpr.2019.05.006.)
- Fish scale valorization by hydrothermal pretreatment followed by enzymatic hydrolysis for gelatin hydrolysate production (Zhang, Y.; Tu, D.; Shen, Q.; Dai, Z. Fish scale valorization by hydrothermal pretreatment followed by enzymatic hydrolysis for gelatin hydrolysate production. Molecules 2019, 24, 2998.)
- The effect of alkaline pretreatment on the biochemical characteristics and fibril-forming abilities of types I and II collagen extracted from bester sturgeon by-products (Meng, D.; Tanaka, H.; Kobayashi, T.; Hatayama, H.; Zhang, X.; Ura, K.; Yunoki, S.; Takagi, Y. The effect of alkaline pretreatment on the biochemical characteristics and fibril-forming abilities of types I and II collagen extracted from bester sturgeon by-products. Int. J. Biol. Macromol. 2019, 131, 572-580, doi:https://doi.org/10.1016/j.ijbiomac.2019.03.091.)
- Effect of acid, alkali and alkali–acid treatment on physicochemical and bioactive properties of hydroxyapatite derived from Catla catla fish scales (Deb, P.; Deoghare, A.B. Effect of acid, alkali and alkali–acid treatment on physicochemical and bioactive properties of hydroxyapatite derived from Catla catla fish scales. Arabian Journal for Science and Engineering 2019, 44, 7479-7490, doi:10.1007/s13369-019-03807-9.)
Thus, the above information from different studies could not be compared with our work. Please kindly understand this limitation.. However, the discussion with the related works but different in raw material used had been mentioned such as in line 184 – 186, 217 – 220, 253 – 257, 308 – 311, 373 – 377, 382 – 385.
- Please increase the DPI values of figures. The quality of them is poor. Also, please make the tables more desirable. They are not looking that much good.
***All of figures are original file with the best resolution we have. Sorry for such a limitation. In addition, the results of color and particle size have been split from Table 1 into Table 2 following the suggestion from another reviewer.
For the tables under the main figures, they are necessary to provide the important value, which cannot be seen or obtained from figures directly. Also, it is better to present figure and details (table format) together. Those figures are listed and the rebuttal is given as follows:
Figure 4 represents X-ray diffraction (XRD) spectra ,while the table provides crystallinity.
Figure 5. FTIR spectra at wavenumbers, while the table provides the wavenumber of different peaks.

Round 2
Reviewer 2 Report
Now, it can be published.